# Coastal flooding will disproportionately impact people on river deltas

Douglas A. Edmonds [1] ✉, Rebecca L. Caldwell[1,5], Eduardo S. Brondizio [2,3] & Sacha M. O. Siani [3,4]

Climate change is intensifying tropical cyclones, accelerating sea-level rise, and increasing coastal flooding. River deltas are especially vulnerable to flooding because of their low elevations and densely populated cities. Yet, we do not know how many people live on deltas and their exposure to flooding. Using a new global dataset, we show that 339 million people lived on river deltas in 2017 and 89% of those people live in the same latitudinal zone as most tropical cyclone activity. We calculate that 41% (31 million) of the global population exposed to tropical cyclone flooding live on deltas, with 92% (28 million) in developing or least developed economies. Furthermore, 80% (25 million) live on sediment-starved deltas, which cannot naturally mitigate flooding through sediment deposition. Given that coastal flooding will only worsen, we must reframe this problem as one that will disproportionately impact people on river deltas, particularly in developing and least-developed economies.

[1] Department of Earth and Atmospheric Sciences, Indiana University, 1001 E. 10th St., Bloomington, IN 47401, USA. [2] Department of Anthropology, Indiana University, 701 E Kirkwood Ave, Bloomington, IN 47405, USA. [3] Center for the Analysis of Social Ecological Landscapes, Indiana University, 701 E Kirkwood Ave, Bloomington, IN 47405, USA. [4] Department of Geography, Indiana University, 701 E Kirkwood Ave, Bloomington, IN 47405, USA. [5] Present address: Chevron Energy Technology Company, Chevron Corporation, 1500 Louisiana St, Houston, TX 77002, USA. ✉email: edmondsd@indiana.edu

People have been exploiting the resources and natural infrastructure of river deltas for at least 7000 years[1]. Most civilizations preferentially grew around coastlines and river deltas because the abundant food resources provided by the sea, the fertile soils, and their positions as transportation hubs fueled development of urban economies and lifestyles[2–4]. This has scarcely changed today as the most densely populated cities in the world are on low-lying deltaic landforms[5,6].

The presence of people on river deltas for millennia and the modification of upstream watersheds have had adverse effects on deltaic landforms[7]. To accommodate the burgeoning populations on the coast, humans engineered rivers[8,9], withdrew subsurface resources[10], and changed the landcover[8]. These changes reduced river sediment supply[9] and increased subsurface subsidence[11], which together initiated erosion and land loss in some major deltas[12–15]. Land surface subsidence locally accelerates relative sea-level change[16] and under available scenarios for sea-level rise, deltaic areas susceptible to coastal flooding could increase by 50% (ref. [13]). Exacerbating these concerns, hydrological extremes are also projected to become more intense, for example tropical cyclones are estimated to be 2–11% more intense by year 2100 (refs. [17,18]). To plan for and mitigate these hazards, we need to know how many people live on deltas and their vulnerability to flood hazards.

An important reason that people living on river deltas are vulnerable to hazards, like flooding, is that physical stressors intersect with multiple socioeconomic and environmental stressors. Deltaic landforms, by their nature, exist at or near sea level. Commonly, low elevation deltaic areas that are prone to storm surge flooding are occupied by low-income residents. These areas can be densely populated, such as the high-density rural areas of the Ganges–Brahmaputra and Mekong deltas[19] or the urban areas of developing and least-developed economies. These low elevation areas also have high levels of infrastructure deficiencies, such as inadequate or nonexistent storm and surface drainage, collection of domestic sewage and trash, and paved roads and/or accessible pathways[20]. On top of that, the inhabitants are experiencing water, soil, and air pollution, poor and subnormal housing infrastructure, and limited access to public services[20]. These stressors undermine both the generic (infrastructural) and specific (individual and group) adaptive capacities of deltaic populations to flood hazards[21,22].

Despite the importance of river deltas as population centers, the estimates of the number of people living on deltas vary widely[23–25]. One reason for this is that there is no widely agreed upon definition of deltaic area and thus there have been few attempts to survey the global deltaic population. Defining delta area is challenging because river deltas are depositional sedimentary bodies that rarely have an identifiable and mappable boundary that defines delta extent. To address these challenges, we developed a new global dataset of delta area to define the global deltaic population, and its vulnerability to flood hazards.

Our analysis shows that in 2017 there were 339 million people living on river deltas. Of these, 329 million (or 97%) were living in developing and least-developed economies as defined by the 2019 UN World Economic Situation and Prospects. Between 2000 and 2017, the global population on river deltas grew by 34% (87 million people), virtually all of it in developing and least-developed countries. We show that geographically, 89% (302 million) of people on river deltas live in the same latitudinal zone as most tropical cyclone activity (5° S to 25° S and 5° N to 35° N)[17]. The 100-year flood from tropical cyclone activity is projected to impact 76 million people across the globe[26], and surprisingly our analysis suggests 41% (or 31 million) of all people exposed to cyclone flooding live on deltas. Of the people on deltas exposed to flooding, 92% (or 28 million) live in developing or least-developed

economies, where lacking infrastructure for hazard mitigation increases their vulnerability. Furthermore, 80% (or 25 million) live on sediment-starved deltas that are unable to naturally mitigate flooding through sediment deposition. Because so many people live on vulnerable, sediment-starved deltas, solutions to mitigating coastal flooding should consider both engineering options and nature-based solutions.

## Results

**Determining global delta population and area.** Our global dataset combines 2174 delta locations[27] with polygons that define delta area. We define delta area as the extent of geomorphic activity created by deltaic channel movement, and delta progradation. We focus on channel network activity because it defines the most flood-prone zone and creates the resources and natural infrastructures that make deltas attractive sites for habitation. We define deltaic polygons with five points that encompass deltaic activity. These five points mark visible traces of deltaic activity with two points capturing the lateral extent of deposition along the shoreline, and with three points enclosing the up and downstream extent of deposition (see Supplementary Figs. 1 and 2, and "Methods" for description of how these points are selected). The convex hull around these five points defines a delta polygon (Supplementary Fig. 3). The total area is calculated by summing areas of all pixels defined as deltaic. Deltaic pixels are defined using an elevation threshold to eliminate high-elevation pixels within that polygon (see "Methods"). For each delta, we report a habitable area (just land, see Methods for more detail) and geomorphic area (land and water). The total population for each delta is calculated by summing the 2017 LandScan population counts[28] of all deltaic pixels within the polygon (see "Methods"). While our method for selecting the delta polygon points introduces some subjectivity, our measured deltaic areas are similar to previous results (see Supplementary Fig. 4 and validation section in "Methods").

**Global distribution of deltaic area and population.** Using our newly defined delta area polygons, we calculate that deltas occupy 0.57% of the earth's land surface area, but they contained 4.5% of the 2017 global population (Fig. 1). Globally, river deltas contain 847,936 km$^2$ of geomorphic area, 710,187 km$^2$ (or 84%) of which is habitable (habitable land is geomorphic area minus water area, see "Methods" for more details) (Fig. 1a). Roughly, 75% of geomorphic and habitable area is found between 9° S and 36° N (Fig. 1c). The largest deltas are the Amazon and the Ganges–Brahmaputra, which contain 84,429, and 80,174 km$^2$ of geomorphic area, respectively (Supplementary Table 1). The ten largest deltas account for 48% of the total geomorphic area (Supplementary Table 1).

Our data show that in 2017 there were 339 million people living on river deltas. People generally do not inhabit deltas at high and low latitudes, and instead 82% of all people living on deltas are in a narrower zone from 9° N to 36° N (Fig. 1c, d). The most populated delta is the Ganges-Brahmaputra with 105 million people, over half of which are in rural areas[19], and the second most populated is the Nile delta at 45 million. In fact, the ten most populated deltas account for 78% of the total deltaic population (Supplementary Table 1).

Deltas are some of the world's most densely populated landforms. For instance, if all 339 million people were evenly distributed across all deltaic habitable area, there would be 478 people/km$^2$ living at a density 8 times the global average. If, on the other hand, we consider the population per delta, then the larger deltas have larger populations, though there is considerable scatter in the relationship especially at small delta size (Fig. 2a).

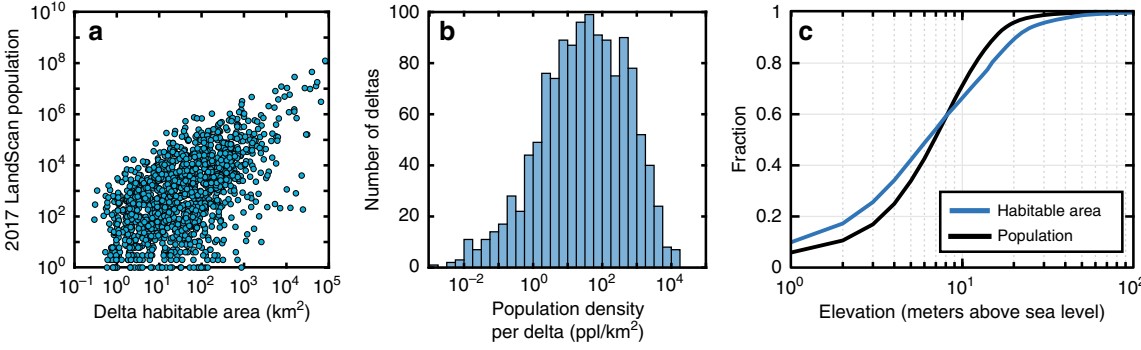

**Fig. 1 Global distribution of deltaic area and population. a**, **b** Total deltaic area and population per 3° lengths of coastline. Lengths of coastlines are colored by the percentage of area or population they contain relative to the entire dataset. Black lines are unmapped shorelines from Caldwell et al.[27]. **c**, **d** Histograms showing the latitudinal distribution (3° bins) of habitable area and population. White bars show the proportion of area and people in the 100-year storm surge floodplain.

**Fig. 2 Statistics of delta area and population. a** Population scales with habitable area. Each dot represents a single delta ($n = 1652$). There are 522 deltas either with no measurable population or habitable area; **b** Histogram of deltaic population density calculated as the total population for each delta relative to the habitable area ($n = 1652$). **c** Cumulative distribution function of habitable area and population as a function of elevation. Distribution only includes deltaic pixels with an elevation of 100 m or less, which accounts for 99.7% and 99.9% of the total habitable area and population, respectively.

The population density per delta, given as the total population on the delta divided by the habitable area of the delta, is log-normally distributed with a median population density of 34 people/km². But our dataset also contains seven highly densely populated deltas with more than 10,000 people/km² (Supplementary Table 1). The Neva River delta in Russia, which contains St. Petersburg, is the most densely populated at 17,062 people/km². The large range in population density (Fig. 2b) arises because

deltas host some of the world's most densely populated cities and, on the other hand, nearly 22% of all deltas ($n = 478$) have equal to or less than 1 person/km² (Fig. 2b).

To put these numbers in perspective we compare them to the area and population of the low elevation coastal zone (LECZ). The LECZ is defined as the land area contiguous with the coastline at or below 10 m elevation[6] and it is often singled out because it is a densely populated zone along the coast that is

experiencing faster than average population growth rates and is vulnerable to coastal flooding[29]. Following other methodologies for the LECZ[6,29], we use year 2000 population counts from Global Rural and Urban Mapping Project[30] for all coastline-contiguous pixels to calculate population and land area within each deltaic polygons that is also part of the LECZ.

We find that even within the fast-growing, highly vulnerable LECZ, there is a clear preference for people to inhabit deltaic landforms. Even though deltas account for 17% (or 445,982 km$^2$) of the global LECZ area, in the year 2000 they contained 32% (or 203 million) of the 625 million people across the globe living in the LECZ[29]. The global population density in the LECZ in year 2000 was 241 ppl/km$^2$ (ref. [29]), though within deltaic parts of the LECZ that number is almost twice as high at 455 ppl/km$^2$.

**Vulnerability of deltaic population to coastal flooding.** Most all people living on river deltas (302 million or 89%) also live in the same latitudinal zone as tropical cyclone genesis in the Northern and Southern hemispheres (5° S to 25° S and 5° N to 35° N)[17] putting them in the path of major coastal storms. As these coastal storms make landfall, they are likely to cause flooding, but the vulnerability of the populations living there depends on both physical and socioeconomic factors. From a physical standpoint, vulnerability to coastal flooding depends on where people live relative to sea-level for a given storm surge height. People are spread out evenly over deltaic elevations; roughly 50% of both deltaic area and population are below or above an elevation of 6.5 m (Fig. 2c). The lowest elevation areas are more prone to flooding and in our data 9.4% of deltaic area and 5.8% of people (or 19.8 million people in 2017) are at or below 1 m elevation (Fig. 2c). Cross-referencing our data with recent global estimates of the 100-year storm surge elevation[26], we find that 11% of habitable deltaic area and 9.1% of all people living on deltas are in the 100-year storm surge floodplain (see "Methods"). Across the globe, 76 million people are exposed to a 100-year storm surge flood[26], and nearly 41% of those people (or 31 million people in 2017) live in river deltas.

Socioeconomic factors also influence vulnerability because they correlate with the quality of physical infrastructure and access to social services, and thus, the ability of deltaic populations to respond to flood risk. Since the 1970s, the pace of urban population growth in developing and least-developed economies has been significantly higher compared to developed ones[19]. The expansion of unplanned settlements in these areas has largely outpaced infrastructure development. Because of this expansion, urban areas in developing and under-developed countries[19,20] have statistically significant lower socioeconomic (e.g., literacy rate, mortality, employment, poverty rate, and quality of life index) and infrastructure (e.g., improved water, percentage slum households, internet access, and city prosperity index) conditions compared to developed countries, both of which directly affect local vulnerability to flooding. For instance, the percentage of slum households in urban areas, which tends to be nonexistent or below 5% in developed economies (such as in North America, Western Europe and Scandinavia, Australia, and Japan), varies from around 10% to over 50% in cities in Asian, Latin American, and African countries that are characterized as developing or least-developed economies. More broadly, a comparative global infrastructural development index (which includes indices for housing infrastructure, social infrastructure, information and communication technology index, and urban mobility) ranges in average from over 80% in developed economies to 40% in Africa, 55% in Asia, and 65% in Latin America. These differences are also reflected in terms of quality of life index[19]. These deficiencies are especially pronounced in high-density rural areas commonly

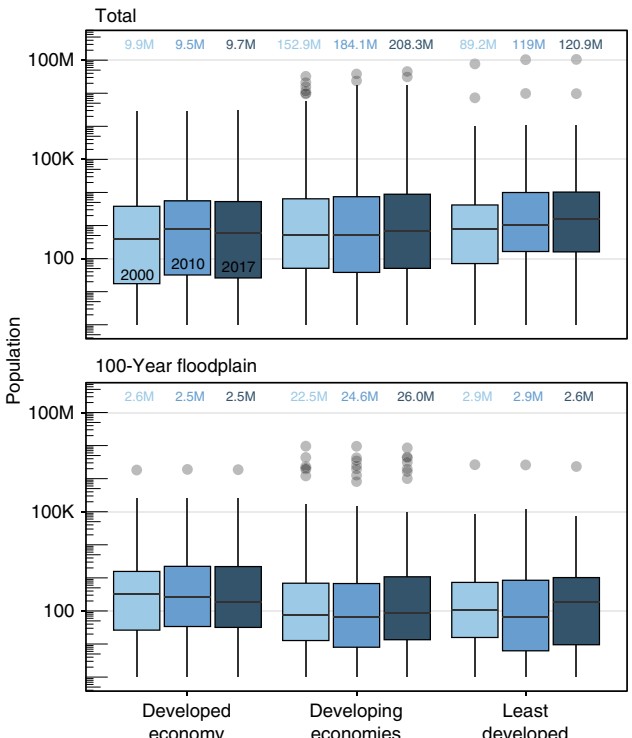

**Fig. 3 Population distribution for deltas classified according to United Nations development categories for their respective countries.** The box and whisker plots show the distribution for all deltas with a given economic development category for each year. The median value is the horizonal line in the box, box width corresponds to the upper and lower quartiles. The whisker lengths represent the lower and upper 25% quartile distribution of all deltas within a category, and gray dots are outliers. Colored numbers refer to the total in each category.

found on deltas, such as the large deltaic populations of the Ganges–Brahmaputra and the Mekong[19].

Developing or least-developed economies with lower socio-economic and infrastructure conditions are more vulnerable to coastal flooding. This is critical because 97% (or 328 million) of all people living in deltas are part of developing or least-developed economies. In 2017, deltas in developing and least-developed countries accounted for 61% (or 207 million people) and 36% (or 121 million people) of the total deltaic population, respectively. These populations are also growing faster than those in developed countries. Between 2000 and 2017, the global population on river deltas grew by 34% (86 million people), virtually all of it in developing and least-developed countries (Fig. 3). Of the people living in the 100-year storm surge floodplain, 92% are in deltas in developing and least-developed countries (Fig. 3).

The 328 million people living on deltas in developing or least-developed economies are not evenly distributed globally. In 2017, for instance, 79% of the total population living in deltas were in the Asia-Pacific regions (259 million people), followed by 19% in Africa (62 million people), 2.9% in the Americas (9.5 million people), and 1.2% in Europe and Central Asia (3.8 million people) (Fig. 1c). Of the deltaic population that lives within developing or least-developed economies and within the 100-year storm surge floodplain, the majority are in the Asia-Pacific region (25 million people), followed by also large populations the Europe-Central Asia (3.8 million people), Africa (3.3 million people), and to a lesser extent, the Americas (~0.5 million people) (Supplementary Fig. 5).

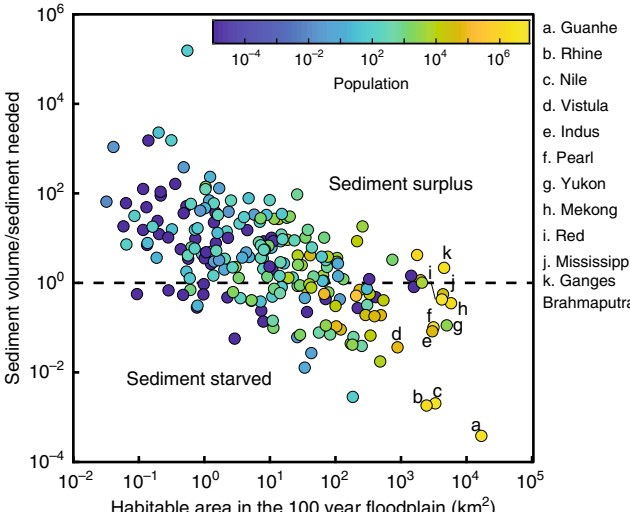

**Fig. 4 Sediment-starved deltas contain more area and people in the 100-year floodplain.** Sediment volume is the depositional volume created by 100 years of river sediment supply[52] with a porosity fraction of 0.4 and assuming 35% of incoming sediment is retained on the delta (ref. [35]). Sediment needed is the volume of space between the 100-year storm surge elevation and the land surface elevation. Deltas with a value greater than one have a sediment surplus and may be able to aggrade their floodplain to the elevation of the 100-year storm surge, and those with value less than one are sediment starved. Each dot represents a delta that has a sediment discharge value ($n = 287$).

## Discussion

Coastal flooding will continue to be a problem for many deltaic communities, making adaptation and mitigation measures a current priority. Some deltaic communities (e.g., the Mississippi, Rhine, Mekong, and Nile) have already adopted engineering solutions, like river management through levees and dams, or land building through diversions[31,32], to mitigate hazards because of the significant flooding risk[7,13]. But, these engineering solutions are expensive and can fail when floods exceed the design limitations. A nature-based solution to limit coastal flooding is to allow deltaic growth to fill in these flood zones with sediment[33–35]. Indeed, this is what a delta does as it grows; areas that are dissected by channels repeatedly flood and receive more sediment[36,37]. In this way, deltas can self-limit flooding if the land surface can aggrade to the elevation of the 100-year storm surge with sediment supplied from the river. Of course, in this scenario we assume that deltas can receive transport sediment where it needs to go in a relatively quick timeframe.

But the bigger issue is that most deltas are at the mouths of the world's major rivers[27] and these rivers often have dams on them, which starve the deltas of sediment[9,13]. We define sediment-starved deltas as those that do not have enough incoming sediment to aggrade their surface to the elevation of the 100-year storm surge event. Comparing the volume of incoming sediment relative to the volume that must be filled, our data show there are two classes of delta. Sediment-starved deltas usually have large floodplain areas (>100 km²) and will not be able to naturally aggrade these flood zones with sediment (Fig. 4). By contrast, sediment-surplus deltas tend to be smaller in area (<100 km²) and theoretically have enough sediment to naturally aggrade their surface and limit coastal flooding, something also noted by Giosan et al.[34] (Fig. 4). But there is an important population difference between these classes. Of the people living in the world's deltaic 100-year storm surge floodplains, nearly 80% (or 25 million) live on sediment-starved deltas. This is critically

important because flood mitigation efforts on sediment-starved deltas will increasingly need to rely on hard engineering solutions because these larger deltas can no longer naturally aggrade their floodplain surfaces, which seemingly limits nature-based solutions.

In sum, our analysis shows that globally people living on river deltas are disproportionately vulnerable to coastal flooding. We show that 339 million people live on deltas, and 302 million of these people live in the latitudinal zone of tropical cyclone genesis. In total, 31 million people live in the 100-year tropical cyclone floodplain. These 31 million people constitute 41% of global population predicted to be at risk from coastal flooding[26]. Furthermore, 25 million (of the 31) live on sediment-starved deltas where nature-based solutions to mitigate coastal flooding will be challenging to implement. Finally, 28 million (of the 31) live in developing of developed economies and lack the infra-structure needed to mitigate flooding.

To effectively prepare for more intense future coastal flooding[11], we need to reframe coastal flooding as a problem that disproportionately impacts people on river deltas in developing and least-developed economies. Reframing the problem is important because river deltas present special challenges for predicting floods. For instance, our estimates of people on deltas exposed to flooding are likely a minimum because global storm surge models[26] currently do not account for compound events created by the interaction of storm surge, rivers, and tides[38,39], and for interactions with deficient urban infrastructure or in areas of high population density. Deltas are also particularly challenging locations to predict storm surge because the distributary channel network, a common feature of most large deltas, allows the surge to propagate inland through the network. Moreover, most coastal elevation models are based on radar data that records the elevation at the top of the vegetation canopy instead of the bare earth elevation. As coastal elevation models improve, it is clear that there are more people at risk of coastal flooding than previously thought[40]. To more accurately assess risk and vulnerability, we need better elevation and storm surge models for deltaic environments[41], as well as finer-grain demographic, socioeconomic, and infrastructure data. Future adaptation and mitigation responses will require models capable of simulating compound flooding[42] in densely populated areas so that exposure and risk can be accurately mapped.

## Methods

**Delta area mapping.** We define the area for each delta identified in Caldwell et al.[27]. Defining delta area is not trivial, and in fact, of the existing studies that report delta area[13,16,23,24,43–47], the methods for defining delta area are not consistent and in many cases are not described. As an example, consider that in two different studies[13,43] the Vistula delta in Poland is listed as 500 km² and 1490 km². Similarly, the area of the Amazon delta is reported as large as 467,000 km² (ref. [43]) or as small as 160,000 km² (ref. [48]). The methods for determining the area in these cases is not clearly explained making it hard to identify the source of the discrepancies. These kinds of discrepancies probably arise because defining the size of any depositional sedimentary body, like river deltas, is difficult because the thickness of deposition usually exponentially declines away from the point source[49]. Tracing exponentially declining deposition to the absolute margin of the deposit can be difficult, if not impossible, because the thickness of sediment deposition becomes vanishingly small. If the thickness of the deposit is perfectly known then one could define a semi-arbitrary boundary for the deposit edge, such as the e-folding length. However, sediment thicknesses for the world's coastlines that distinguish deltaic and nondeltaic deposition are not easily obtainable and defining delta size based on deposit thickness is not feasible. Instead, we think that the most reliable data we can use to define delta area are high-resolution (15 cm to 15 m) images available in Google Earth. Even from imagery, the extent of delta deposition is difficult to measure because it may interfinger with adjacent coastal environments creating a gradual transition that is difficult to identify on a high-resolution image. Of course, in some cases this may not be true, because if deposition is confined, within a valley for example, then the contact between deltaic and nondeltaic area can be mapped with confidence (Supplementary Fig. 3a, b).

But not all deltas form in valleys or places where their lateral contacts are visible, so this criterion cannot be universally applied.

Considering these challenges, we define delta area as all land where deltaic processes are visibly active now or and in the past. The method we use to define delta area relies only on surficial information that is visible in images. We adopted a simplified approach using five points to define area because it can be applied to every delta and only requires an image to implement. We mark visible traces of deltaic activity with two points capturing the lateral extent of deposition along the shoreline (S1 and S2), and with three points enclosing the up and downstream extent of deposition (RM, OB, DN) (Supplementary Figs. 1, 2; Supplementary Table 1). The convex hull around these five points defines a delta area polygon (Supplementary Figs. 1c and 3). Detailed definitions of these points are provided in the following section.

Our method captures the first-order shape of a delta with operational definitions that are straightforward to apply. Admittedly, this approximation does not perfectly capture all intricacies of delta shape (Supplementary Fig. 3). While these choices introduce some subjectivity, this method is consistent with previously measured delta areas (Supplementary Fig. 4 and Table 2; see validation section of "Methods"). We provide all our point selections so that individual decisions can be assessed on a case-by-case basis (see Supplementary Data).

**Considerations for locating delta extent points.** The locations of the five points that define delta area were chosen using the most recent imagery available in Google Earth. Due to the rapidly changing nature of deltaic land and updated imagery in Google Earth, some of these point locations will likely change with time, and may differ from the points we define at the time of this paper.

*River mouth (RM)*: On each delta we marked the location of the widest river mouth in the distributary network at the shoreline. We choose the widest channel because that is the one likely carrying the most water and sediment.

*Delta node (DN)*: The DN is defined as either (1) the upstream-most bifurcation of the parent channel (Supplementary Fig. 2a), or if no bifurcation is present, as (2) the intersection of the main channel with the delta shoreline vector ($L_S$) which is defined as the line connecting S1 and S2 (Supplementary Fig. 2b). In the case where both (1) and (2) exist, the DN that is furthest upstream is chosen as the DN location. If a delta does not have a distributary network, then option (2) is chosen as the DN.

*Lateral shoreline extent points (S1 and S2)*: The lateral shoreline extent points are defined as either (1) the locations on the shoreline that mark the boundary between deltaic protrusion and the regional nondeltaic shoreline (Supplementary Fig. 3c), or (2) the lateral-most extent of channel activity, defined by an active or inactive channel (Supplementary Fig. 1a). If both (1) and (2) exist, the lateral shoreline extent locations that are farthest laterally from the center of the delta sets the S1 and S2 locations. Point S1 is on the left side looking upstream, and point S2 is on the right side looking upstream. When considering criteria (1), finding an obvious boundary between deltaic protrusion and the regional nondeltaic shoreline is not trivial, because deltaic deposition declines exponentially away from the source. In simple cases, such as wave-dominated cuspate deltas, the shoreline extents correspond to the maximum curvature of the delta shoreline protrusion as it transitions to the regional shoreline trend (Supplementary Fig. 3c). In non-obvious cases, we aim to select the location that marks a transitional zone between deltaic and nondeltaic, and because of this, individual points may have different interpretations. In some more complicated cases, deltas can merge at the shoreline and may share a point (Supplementary Fig. 1c).

*Basinward extent point, toward open basin (OB)*: This point is defined by the location of delta land (not including islands detached from the shore) that is farthest basinward measured perpendicular to the delta shoreline vector ($L_S$) (Supplementary Fig. 2).

In addition, we considered the following. Channels that are both active and inactive in Google Earth imagery (i.e., holding water or not) were used for determining any of the above point locations that may be distinguished by the location of a channel body (i.e., DN, S1, S2) (Supplementary Fig. 1a, channel on right demarcated by light blue arrow). We include inactive channels because they are evidence of deltaic deposition and there is no way to conclude if they are only temporarily inactive at the time the image was captured. Examples of inactive channels include temporarily inactive channels, such as ephemeral rivers or tidal channels, as well as channels that have been abandoned through avulsion but are still distinguishable in aerial imagery. For example, a delta's node may be chosen by an avulsion point of the parent channel creating a network of both currently active and inactive distributary channels downstream (e.g., Supplementary Fig. 1a).

In addition, obviously human-made channels/canals were not included when defining the lateral extent of a channel network. We visually judged channels to be human made based on their straightness. But natural channels are often artificially stabilized by human activities, and we use these channels to define the delta extent when they could be clearly traced upstream to a natural channel (Supplementary Fig. 1b).

Multiple rivers can interact to form one delta (e.g., one clear continuous protrusion from the shoreline). These multiple-source deltas are represented by one entry in the dataset (Supplementary Fig. 1c, blue arrow indicates second river forming ID: 4023 on right). If two rivers create two deltas that are next to each other with some distributary overlap, they are represented by two entries in the

dataset (Supplementary Fig. 1c). Transitional cases are common, and thus the distinction between these two cases is not always clear. When possible, the existence or absence of separate shoreline protrusions were used to determine if multiple proximal rivers are creating one large delta or several slightly overlapping deltas. If two or more rivers overlap via small tidal channels or human-made canals, they are not considered to be 'interacting' and are marked as separate entries in the dataset.

**Calculating delta geomorphic area and habitable area.** We calculate two area values: geomorphic and habitable. In both cases, we first remove land that falls within the delta extent polygon that is much higher elevation than the surrounding delta plain. High topography may be included inside a delta polygon when deltaic deposition fills in areas between pre-existing high topography. For example, this occurred in the Acheloos delta (Greece)[50]. To objectively remove high-elevation nondeltaic areas for both the geomorphic and habitable area, we define elevation outliers as those points that are more than two times the inner quartile range of the elevation data for a given delta. Based on inspection, this effectively removes high-elevation nondeltaic areas that are included in our delta polygons. Along the boundaries of the polygons we included the pixels if more than 50% of the pixel area was inside the polygon. Once clearly nondeltaic land is removed, we calculate the geomorphic area as the cumulative sum of all remaining pixels within the polygon. This area can include channels, shallow marine zones, and other bodies of water that are included in the polygon (Supplementary Fig. 3).

Habitable area corresponds to the amount of land—geomorphic area minus the cumulative water (both fresh and saline) area—within each delta polygon. We call this habitable area under the assumption that people would not find water environments suitable for habitation, and only rarely live permanently on the water, although in some delta sectors people live on stilt habitations above the water. The land and water proportion for each pixel is determined at a subpixel level from a water mask that defines locations of water bodies like channels, wetlands, lakes, and the ocean. For this proportion we used a publicly available raster dataset of land and water area per pixel[7]. Pixel size is 30 arc second, or 1 km at the equator. Total habitable area is then the sum of all these proportions that fall within the polygon and are not masked out by the high-elevation criterion.

Because some deltas are smaller than the 30 arc second pixel size, not all deltas in the database have a geomorphic or habitable area value ($n = 522$) and were given a value of NaN.

**Delta area sensitivity and validation.** Our methodology draws a hard boundary by separating deltaic from nondeltaic land. This boundary is geomorphically defined, and population centers may straddle this boundary or lie just outside of it. A softer approach that also counts the population near the delta polygons may yield different estimates. To assess the sensitivity of our results to our choices of delta extent points, we create new polygons that are twice as large as the original by isotropically dilating the shape. This way we can also capture the population immediately adjacent to deltas. When we use these dilated polygons, we calculate a new global delta habitable area of 1,060,000 km² and population of 522 million. The population increases, as expected, but the population density stays relatively constant (492 ppl/km² instead of 478 ppl/km²). This suggest to us that we are not missing any major, densely populated areas adjacent to our delta polygons. Note that even though we doubled the delta polygon area in this sensitivity test, habitable area did not double (increased from 710,179 to 1,060,000 km²) because it is always smaller than the polygon area because it does not include water bodies.

To validate our delta area methodology, we compare our area measurements based on the five points to delta areas reported by other authors. Even though we find it difficult to assess how other authors measured delta area, this allows us to understand if our measurements capture the spirit of what others tried to do. We cross-referenced our area data with that from Syvitski and Saito[43] and found that our delta area measurements are remarkably close to theirs (Supplementary Fig. 4). In fact, the best fit linear regression nearly has a slope of 1:1 representing minimal bias, and the $R^2$ is 0.91. However, some of the measurements are significantly different than ours. In most cases this occurs because we use active and inactive channels to define the DN and shoreline extents. If we just use active channels (those with water in recent imagery), then our areas for three deltas (Brazos, Niger, and Yukon) are revised downward and come much closer to previously published values (Supplementary Fig. 4 and Supplementary Table 2). The only measurement that is still significantly different is that for the Amazon delta. We report an area of 85,667 km² and Syvitski and Saito[43] report an area of 467,100 km². Recent work by Brondizio et al.[48] suggests that the difference may be because the larger area includes the full extent of tidal channel activity not directly connected to the main river and channel network. In Brondizio et al.[48], the Amazon delta area was defined as a social–ecological system based on the intersection of physical and political administrative and demographic units, and this led to an estimated area of 160,662 km². Given the large uncertainty in the area of the Amazon delta, we show it on Supplementary Fig. 4, but do not include it in the linear regression. The average percent error between our measurements and Syvitski and Saito (excluding the Amazon) is 50% and if we use the revised areas for the three deltas (Brazos, Niger, and Yukon) the average percent error is 36% (Supplementary Table 2).

**Calculating delta population and designating country development categories**. To generate population counts for each delta polygon, we use the Oak Ridge National Laboratory LandScan dataset[28]. We choose LandScan because it is based on census data, and uses a multivariable dasymetric model and imagery analysis, including nightlights, to spatially disaggregate the population. This is critical because it more accurately reflects the population at the coastline. For instance, other population datasets, like Global Population of the World (GPWv4)[7], spatially distribute all population within a given administrative boundary. This can lead to an overestimation of population at the coastline, when for example the population of a nearby city is distributed to the coast even when few people live there (Supplementary Fig. 6). In addition, LandScan extends all coastal boundaries several kilometers seaward to capture the people living along the shoreline. For those reasons we prefer the LandScan dataset.

The population for each delta area polygon was calculated by summing all the pixels of the population raster that fall within the delta extent polygon. Like the area calculations, pixels on the border were included if more than 50% of the area was inside the extent boundary. Because some deltas are smaller than the 30 arc second pixel size, not all deltas in the database have a population value. There were 549 deltas that were given a value of NaN for population. These were excluded from the analysis.

We also compared our LandScan-derived population numbers to Global Population of the World (GPWv4)[7] and GRUMPv1 (refs. [7,30]). The GPWv4 dataset spatially disaggregates and rasterizes the population within a given geographic boundary using a uniform areal-weighting method. Population data come from census tables. The GRUMPv1 population dataset is based on georeferenced census data that are allocated to urban and rural areas. With the GPWv4 dataset, we calculate a total global delta population of 360 million people in 2020. GRUMPv1 is only available for year 2000 and we calculated a total population of 269 million, which is similar to the 252 million calculated for LandScan for that year. The picture is similar if we consider the population within the 100-year floodplain for these different datasets. Using GRUMPv1 (year 2000) and GPWv4 (year 2020) we calculate that 37.8, 42.8 million people, respectively, reside in the 100-year floodplain.

Using a country boundary map as overlay, each delta was associated with a country, which in turn was designated to an economic development category based on the 2019 United Nations World Economic Situation and Prospects. Three categories are used: developed, developing, and least-developed countries. In addition, for the purpose of regional comparisons, we used country designation to assign each delta one of four global regions (Asia-Pacific, Americas, Europe-Central Asia, and Africa), as defined by the United Nations, as shown in Supplementary Fig. 5.

**Calculating 100-year floodplain area and storm surge elevation**. The 100-year floodplain area is calculated as the area at or below the elevation of the 100-year storm surge that is also connected to the ocean, either directly or via a river channel. Pixels are considered connected if any of the eight surrounding pixels have an elevation below the storm surge value. The elevation of the 100-year storm surge for each delta is determined by using the median value of all storm surge values calculated by Muis et al.[26] that fall within the delta polygon. This analysis does not account for the presence of coastal flood defenses. For instance, the Rhine delta has a high population within the 100-year floodplain, but their vulnerability is lower than less developed deltas. We use the Muis et al.[26] study of Global Tide and Storm Surge Reanalysis (GTSR) to estimate 100-year storm surge elevation because it is based on hydrodynamic modeling and has been rigorously validated by comparing modeled and observed sea levels. This approach is termed 'dynamic' because it simulates the creation and propagation of the storm surge in a hydrodynamic model. Other models for storm surge, for instance the DINAS-COAST Extreme Sea Levels (DCESL), rely on static approximations of storm surge conditions and mean high tide. Comparison of these methods shows that DCESL overestimates extremes by 0.6 m, whereas GTSR underestimates by −0.2 m[51].

**Calculating delta elevation**. For all calculations involving elevation we use the Global Multi-resolution Terrain Elevation data from 2010 courtesy of the USGS. This composite dataset consists of elevation data from multiple sources. Because the data come from multiple sources the native resolution is not consistent and the raster has to be aggregated to a consistent resolution. We use an aggregate raster that reports the mean elevation of the native data at a resolution of 30 arc second.

In a recent publication, Muis et al.[51] pointed out that the datum for GTSR is mean sea level and that is not the same for the elevation data used here (EGM96). We opt to not correct the datum for GTSR so that we can make a direct comparison with Muis et al.[26].

**Reporting summary**. Further information on research design is available in the Nature Research Reporting Summary linked to this article.

## Data availability
All data generated and analysed during this study are included in this published article (and its supplementary information files).

## Code availability
Code needed to recreate the datasets used in this work is available in a public repository located at http://hdl.handle.net/2022/25788.

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

## Acknowledgements
This research was supported by National Science Foundation award EAR 1812019, 1426997, 1135427, and support from the Prepared for Environmental Change initiative at Indiana University, awarded to D.A.E. The authors wish to acknowledge the support of Indiana University's Institute for Advanced Studies and Indiana University Library's Open Access Article Publishing Fund.

## Author contributions
D.A.E., R.L.C., and E.B. conceived of the study. D.A.E. executed the study and wrote the initial draft. R.L.C. performed the deltaic area mapping and assisted with data analysis. E.B. and S.S. performed the socioeconomic data analysis. All authors discussed the results and contributed to writing the manuscript.

## Competing interests
The authors declare no competing interests.
