## [Peer Review File · Nature Communications]

Reviewers' comments:

Reviewer #1 (Remarks to the Author):

General comments:

1. This is a good effort on the estimation of the delta population, Throughout the text, the use of secondary and primary source data is a bit confusing. It's better to indicate which are primary and which are secondary source data.
2. There is no mention of image specification if any used, no mention of Softwares used for mapping, no mention of proper sources for the dataset used for this study.
3. For the delta population estimation, bases are not clear which mention in the comment section later.
4. Add some recommendations.
5. The overall arrangement of the manuscript is not eye-catching, the result section comes first, lastly method section and again in between mitigation part. Consider rearranging if possible with a journal format.

Specific comments:

Line 14, Any name of this dataset?

Line 19, no such or very little saying are there in the main text

Page 2 line 33 is there any chance to include river hydrodynamic changes?

Page 2 line 36-37 give some statistics

Page 38, line 38 preferably change the word

Page 2, line 49, put in the different method section. It is a bit confusing to read the analytical methods in between starting paragraphs.

Page 3, 59- 61, basis of taken points, the gap between those points and reason.

Page 3, line 69, not clear the process of identification of deltaic population, have you used any administrative boundary-based population data? Suggest doing so.

Page 4, line 74-75, rephrase or delete

Line 78 May compare these values with some referenced statistics

Line 83 Is it also a result?

Line 90, it's confusing which are primary data sources and which are secondary. Please mention.

Line 109, 110, please explain

Line 111, refer few others studies from vulnerable deltas like GBM delta

Line 112- 115, try to give some actual statistics on these variables from developing / underdeveloped countries

Line 130, is this included in the recommendation part?

Line 134, explain the term

Line 137- 138, tidal channels also play an important role in delta sustainability

Line 151-152, Is it referred?

Line 166, is this method explained in the text? Not sure what it does implies with a number of deltas rather than the location!

Line 170, mention the categories

Line 196, What type of photographs have been used?

Line 196, Is it only google earth image that has been used? What is the specification of images?

Line 198, how would you transfer a photo to the map?

Line 202, Have you used any software for delineating the areas?

Line 204- 207, is there any established basis for the selection of these 5 points?

Line 215, Please mention a few bases of choosing these points.

Line 240 Put a gap

Line 244, specification of imageries

Line 252, How does it differentiate? Through visual interpretation or classifying the image?

Line 270, What elevation data have been used?

Line 315, This process is not clear.

Have you used any administrative boundary to estimate the population?

Often large urban/rural unit does not fully cover with delta areas, how do you delineate the delta population and rest population for those units?

Have you done any survey or use any survey data?

If there any statistical estimation method, kindly mention.

Line 325-326, give some details of this dataset

Line 337, the elevation is not a sole factor behind coastal flooding

Do you use the SRTM image to get elevation values?

Line 343, put a sentence for this model

Line 345, ??

Line 474, is it from the result section?

Reviewer #2 (Remarks to the Author):

This manuscript presents the results of a detailed population-distribution and flood risk assessment of global deltas. The authors build on recent work identifying thousands of global deltas, and develop a new methodology to map deltas and compute their areas in a more rigorous and well-defined manner than previous methods. The manuscript is well-written and would be a valuable addition to the literature, scaling up and adding rigor to previous efforts mapping delta population and risk over smaller samples and with more ad-hoc mapping methods.

Comments:

1. The explanation of how you define delta area extent (briefly in the main text, and more explicitly in the supplement) would benefit from additional specific examples that illustrate the different cases described in the text. You do present several examples, but an example for each "branch" in the decision tree would be helpful, or a flow diagram might help. As it is, the different cases and how they are handled seem somewhat arbitrary, though I don't doubt that there are reasonable explanations for why each special case is needed.
2. Can you comment on how this area method works for river-dominated deltas such as the Mississippi? I would think the convex-hull would grossly overestimate the land area of these systems, at least more so than for wave- or tidal-dominated systems? An example of one of these deltas would be helpful. Or, does this not matter because the excess of water within the polygon in these systems is adjusted when you compute your geomorphic and habitable area estimates?
3. A brief note about how the database of 2000+ deltas were identified by Caldwell 2019 would be helpful.
4. That 41% of the people exposed to 100-year storm surge flooding live in deltas (while only 4.5% of the total population) is striking. How does the total coastal delta area compare with the total Low Elevation Coastal Zone area, and their respective populations? I'm curious if people in deltas are at higher exposure when compared with people in similarly low-elevation non-delta areas. Is the elevated exposure you find due to low elevation itself, or is it associated with deltas specifically.
5. Figure 2 and 3 are interesting, but they aren't discussed much in the main text - we mostly get a set of top-level descriptive statistics. I understand that space in the main text is limited, but are there specific patterns or features in these data distributions that could be highlighted?

Zachary Tessler

Response To Reviewer #2 (our responses to the numbered reviewer comments are *italicized*)

1. The explanation of how you define delta area extent (briefly in the main text, and more explicitly in the supplement) would benefit from additional specific examples that illustrate the different cases described in the text. You do present several examples, but an example for each “branch” in the decision tree would be helpful, or a flow diagram might help. As it is, the different cases and how they are handled seem somewhat arbitrary, though I don’t doubt that there are reasonable explanations for why each special case is needed.

This is a fair suggestion and something we considered when writing the manuscript. But, we don’t feel this would be the clearest way to communicate the information because a decision tree would be needed for each deltaic extent point. This would result in five different decision trees. Besides that, there is not really a definite decision tree that can be made for each deltaic extent point.

2. Can you comment on how this area method works for river-dominated deltas such as the Mississippi? I would think the convex-hull would grossly overestimate the land area of these systems, at least more so than for wave- or tidal-dominated systems? An example of one of these deltas would be helpful. Or, does this not matter because the excess of water within the polygon in these systems is adjusted when you compute your geomorphic and habitable area estimates?

Great question. We think this area measurement works just as well for river-dominated deltas as any other type. Though, we note, there is no easy or straightforward way to classify a delta as river-dominated. The Mississippi River is included in our validation dataset (Supp. Table 2 and Supp. Figure 4). We estimate it at 51,124 km², where Syvitski and Saito report it as 38,568 km², which means we overestimate it by 32.56%. But, the value we report is the geomorphic area, which as the reviewer points out, includes the shallow ocean in front of the delta due to the convex hull we fit around the deltaic points. When we compare the habitable area of the Mississippi River, which has the ocean pixels removed, it is much closer with a value of 39,829 km² (Supp. Table 1). The fact that the convex hull and the geomorphic area leads to an overestimation is not surprising and this is likely to be a bigger problem for the river-dominated deltas that protrude out in the ocean. But, it is worth pointing out that in almost all of the statistics we report we use the habitable area, which does not include the ocean and thus minimizes this overestimation. The use of the habitable area minimizes the error associated with fitting a convex hull around the five deltaic points.

3. A brief note about how the database of 2000+ deltas were identified by Caldwell 2019 would be helpful.

Our methods are already at the limit suggested by Nature Communications. Because of this, we prefer not to provide this detail since it is readily available in an open source journal (Earth Surface Dynamics).

4. That 41% of the people exposed to 100-year storm surge flooding live in deltas (while only 4.5% of the total population) is striking. How does the total coastal delta area compare with the total Low Elevation Coastal Zone area, and their respective populations? I'm curious if people in deltas are at higher exposure when compared with people in similarly low-elevation non-delta areas. Is the elevated exposure you find due to low elevation itself, or is it associated with deltas specifically.

This is a good suggestion and we thank the reviewer for it. We now added a paragraph that compares the portion of the deltaic population that resides in the LECZ to the global LECZ. We found that nearly 1/3 of the global LECZ population lives on deltas at a population density twice that of the LECZ. Please note that it was only possible to do this calculation using data from 2000, which indicates a pattern of distribution that we believe largely applies today.

5. Figure 2 and 3 are interesting, but they aren't discussed much in the main text - we mostly get a set of top-level descriptive statistics. I understand that space in the main text is limited, but are there specific patterns or features in these data distributions that could be highlighted?

*We thank the reviewer for giving us a chance to more fully explore our data. We now discuss these figures more in the text, while still privileging high-level statistics since we want to cover these in this first paper. This issue is now covered in more detail in lines 99-123. We also explore more detailed sub-regional data in **Extended Data Figure 5**. We added additional text on the topic in lines 138-156*

REVIEWERS' COMMENTS:

Reviewer #1 (Remarks to the Author):

The response from the author is satisfactory and well composed. I have few comments overall-

1. Its not very clear to me what method author has took to delineate the delta population. By clipping polygon how does exact population number can be calculated?
 2. Again the image processing tools and techniques are not well explained.
 3. They are using Google Earth images for this study, few refernces on using Google earth on this kind of study would help to establish the method.
- Rest seems fine to me.

I have gone through the author's response to Reviewer 2. The main problem is author doest does not mention the exact section/ para/ lines of their address in the text, so a bit difficult to follow. Mostly I think they have addressed well. Again few lines on 'define delta area extent' would be great.

Response to reviewers on *Coastal Flooding Disproportionately Impacts People in River Deltas*

Our responses are **bolded**.

Reviewer 1

The response from the author is satisfactory and well composed. I have few comments overall-

1. Its not very clear to me what method author has took to delineate the delta population. By clipping polygon how does exact population number can be calculated?

We state on lines 83-88: “The convex hull around these five points defines a delta polygon (Supplementary Data Figure 3). We then use an elevation threshold to eliminate pixels within that polygon that are not deltaic. The total area is calculated by summing all pixel areas, and we report a habitable area (habitable land is geomorphic area minus water area, see Methods for more detail) and geomorphic area (land and water). The total population is calculated by summing the 2017 LandScan population counts²⁷ of all deltaic pixels within the polygon (see Methods).” We rewrote this section to be more explicit about our method. We use the delta polygon and find all raster cells contained within that polygon, and then sum the population count of each cell to get a total population.

2. Again the image processing tools and techniques are not well explained.

We regret that that the methods are not clearer to the reviewer, this may be in part due to some confusion from the reviewer. There are no image processing techniques involved in this research. As we explain, we look at the images in google earth.

3. They are using Google Earth images for this study, few references on using Google earth on this kind of study would help to establish the method.

Google Earth is a standard tool without any obvious scientific citation. We are just using Google Earth for the images, which, in our opinion, does not require citation.

Rest seems fine to me.

I have gone through the author's response to Reviewer 2. The main problem is author does not mention the exact section/ para/ lines of their address in the text, so a bit difficult to follow. Mostly I think they have addressed well. Again few lines on 'define delta area extent' would be great.

We have defined our definition of delta area extent in the methods section in detail.